# Proposed Protocol for Field Testing of Endurance Fitness of Young Labrador Retrievers

**DOI:** 10.3390/mps6040061

**Published:** 2023-06-28

**Authors:** Ella-Erika Söderlund, Heikki Kyröläinen, Outi M. Laitinen-Vapaavuori, Heli K. Hyytiäinen

**Affiliations:** 1Department of Clinical Equine and Small Animal Medicine, Faculty of Veterinary Medicine, University of Helsinki, 00014 Helsinki, Finland; ella.soderlund@icloud.com (E.-E.S.); outi.vapaavuori@helsinki.fi (O.M.L.-V.); 2Neuromuscular Research Center, Faculty of Sport and Health Sciences, University of Jyväskylä, 40014 Jyväskylä, Finland; heikki.kyrolainen@jyu.fi

**Keywords:** physical fitness, fitness test, aerobic fitness, endurance test, dog

## Abstract

The number of dogs and, with it, dog sports are growing in popularity, and the training of dogs begins at an early age. Although fitness testing is an imperative part of purposeful training and sports, to our knowledge, no objective field tests are available for measuring young dogs’ endurance fitness. The aim of this study is to describe a simple, easy-to-repeat, and inexpensive way to test training intervention effects on endurance fitness in young Labrador Retrievers. Healthy client-owned 16-week-old Labrador Retrievers will be recruited and divided into test and control groups. The test group will have an eight-week training program followed by a four-week detraining period, while the control group will live a normal puppy life. All dogs will be tested for endurance fitness four times at four-week intervals: at baseline, one month later, two months later at the end of the training period, and one month after ending the training program. Each of the four testing sessions will be identical and will consist of four measurements of heart rate (HR) and blood lactate (BL): at baseline, after trotting 1000 m, after sprinting 200 m, and at recovery 5–8 min after the sprint. The training-induced changes in endurance fitness are evaluated by changes in HR and heart rate recovery times (HRR), BL, and running times.

## 1. Introduction

In humans, physical fitness is defined by the American College for Sport Medicine (ASCM) as “a set of attributes or characteristics that people have or achieve that relates to the ability to perform physical activity” [1]. Physical fitness is further divided into skill- and health-related components. One of the health-related components is cardiorespiratory endurance, also known as aerobic fitness or endurance fitness [2]. Endurance fitness refers to the capacity of the circulatory and respiratory systems to administer oxygen to skeletal muscles, thus enabling them to produce energy during physical activity [3,4]. When there is insufficient oxygen available, anaerobic processes (glycolysis) occur to allow for energy production, and lactic acid is formed [5]. Accumulation of lactate leads to acidosis, which in turn causes muscle fatigue, attenuating muscle actions, and force production [6].

Aerobic training has been shown to decrease resting heart rate (HR) in humans and horses [7,8,9]. HR monitoring is used to assess the effectiveness of training programs and the level of cardiovascular fitness, as well as to estimate indirect maximal oxygen uptake (VO2max) during exercise and to measure the intensity of physical performance in humans, horses, and dogs [7,10,11,12,13,14,15,16,17]. HR data can be compiled with an electrocardiogram by auscultation at rest or with wearable HR sensors [18]. In field tests, different HR sensor belts are used to measure HR during physical performance [16,19,20]. There are HR sensors and sensor belts designed specifically for horses; for example, Polar Electro Oy (Kempele, Finland) has developed HR sensors suitable for use under the saddle while riding and a handheld sensor for quick HR monitoring. To the authors’ knowledge, there are no HR sensors designed for dogs, but Essner et al. [20,21] have validated one human HR sensor model in dogs.

In humans and horses, heart rate recovery time (HRR) has been thought to indicate changes in fitness level and training status [22,23,24,25,26]. In elite human athletes, HRR is only related to aerobic fitness during the first 10–30 s after performance [25]. In non-athlete humans, HRR has been found to decrease significantly after 8 weeks of training but to return to almost baseline levels after 2 weeks of detraining [24]. Racing horses with lower resting HR before exercise has been shown to have faster HRR [27], which, when noted after interval exercise, is related to better race performance [26]. Nevertheless, there is a lack of information available regarding dogs’ HRR and its relation to their performance capabilities.

One key marker of endurance fitness in humans is the concentration of lactate in the blood relative to exercise intensity [2,28,29]. During exercise, the point where the formation of lactate exceeds the elimination of lactate from the blood is called the lactate threshold or anaerobic threshold [2,29]. Physical performance below this point is considered aerobic exercise [2,29]. Analysis of blood lactate (BL) can also reflect exercise intensity [2,30]. Therefore, BL analysis is routinely performed in human sports medicine [31]. BL analyses are also used in dogs and horses [19,32,33,34,35]. For example, endurance-trained dogs have significantly lower BL concentrations after long-distance running than untrained dogs [36]. Endurance-trained dogs have been found to reach lactate threshold at speeds of 13.9 km/h (±2.9 km/h) [35]. A portable, minimally invasive BL measurement device is readily available for dogs, and it has been shown to be reliable in measuring their BL levels [37]. In this protocol, this method will be used for young Labrador Retrievers.

In addition to the above-mentioned methods, endurance fitness can be tested in humans with several validated field test batteries [38,39,40,41,42,43]. One of the most widely used field tests is the 20-min shuttle run test (20 mSRT) [39,44]. In this test, the participants run at an increasing pace for 20 m back and forth until they can no longer keep up with the demanded speed. This test has been shown to be valid and reliable for measuring cardiorespiratory fitness in both adults and children [39,43,45,46,47]. There are also time-limited tests, such as the 6-min walking test (6 MWT) and the 12-min running test [48,49], in which a distance traveled over a certain period of time is measured. These two tests are not reliable and valid for children, and thus, 20 mSRT should be used [50]. Another option is the distance-limited test, which measures the time spent traveling a certain distance [48,50].

A dog with good physical fitness has a better capability to perform its task, which also reduces the risk of injury; hence, the ability to measure the dog’s fitness level is imperative. With the increasing number of pets and working dogs, owners’ interest in the well-being of their dogs is also on the rise [51,52,53,54,55]. Providing fitness-related information to dog owners can affect their dog handling and exercising habits and thereby contribute to the well-being of dogs. Moreover, the popularity of dog sports is growing, and dogs’ training begins at an early age. This may lead to an increasing rate of musculoskeletal injuries if the dogs’ physics is not appropriately tested and trained.

Field tests for assessing endurance fitness in dogs are scarce. The six MWT is used in non-healthy dogs to characterize the severity of respiratory and cardiac diseases [6,15,56] and to characterize golden retriever muscular dystrophy [57]. The same test has been applied to search-and-rescue dogs too [58]. However, the test does not apply to endurance fitness testing. Another fitness test for assessing dogs’ physical fitness has been published [59], but this test does not include an objective measurement of endurance fitness; instead, it concentrates on various aspects of locomotion.

Despite fitness testing being an important part of purposeful training and sports in other species, there are currently no field tests available for testing young dogs’ endurance fitness. In veterinary medicine, current field tests for measuring endurance fitness are designed for horses [8,9,16,60].

The aim of this study is thus to design a simple, easy-to-repeat, and inexpensive field test to measure the effects of a training intervention on endurance fitness in young dogs. This protocol will not only be useful to veterinary medicine professionals but is also suitable to be used by non-professionals (i.e., owners and dog trainers).

## 2. Procedure

Privately-owned Labrador Retriever sibling pairs aged 16 weeks with no known health issues and with a healthy genetic background based on the Finnish Kennel Club health records will be recruited. Exclusion criteria will be any detected abnormalities in lameness assessment or in the general examination. These assessments will be repeated at each test session right before physical performance. In addition, the dogs will be under the veterinarian’s supervision at all times. All dog owners will sign a written informed consent form before participating in the study. Owners will be advised that they may cancel their participation at any time. Furthermore, the veterinarian can withdraw the dog for any justified reason (i.e., health-related concerns, owner-related behavioral issues, etc.). In the event of withdrawal, any research data collected up to that point will be used. Approval for the study protocol will be sought from the Regional State Administrative Agency for Southern Finland. The dogs will be divided into two groups: from each sibling pair, one of the siblings will be in the training group, and the other in the control group. The dogs will be randomly assigned to these groups. The training group will have an eight-week training program, including exercises on an underwater treadmill and a home exercise program, and a detraining period of four weeks after finishing the training program. The control group will live a normal puppy life, i.e., puppy management will be conducted according to each individual owner’s preferences, with no restrictions or added exercises related to research. Both groups will undergo the same outcome measurements at the same time points (described below), and for both groups, the study period will last a total of 12 weeks.

### 2.1. Study Protocol

The test will be performed in an indoor hall with a controlled environment and temperature (15–20 °C) and on a good, non-slip surface (Agility Softex, Lappset, Helsinki, Finland) to ensure a standardized testing environment. Prior to each test session, the dog will be taken out for a calm walk as a warm-up and to allow it to defecate and urinate. The dog will be allowed to acclimate to its surroundings for 5–15 min before the testing. The length of the acclimation will be based on clinical experience. The time of a test session will remain consistent throughout the study. The measurements of the sibling pair will be conducted consecutively on the same day.

The test protocol is performed in the same way each time (Figure 1). A clinical examination and lameness evaluation will be conducted prior to every test session, right before performance. Lameness evaluation will be conducted by palpation and manipulation as well as gait observation for any indications of unsoundness. Each of the fitness test sessions will consist of two physical performance sets and four measurements of HR and BL (Figure 2). The physical performance will be evaluated by trotting for 1000 m and a sprint test for 200 m. During the “trot” phase, a running trot will be utilized, defined with a running gait with a suspension phase between each diagonal support phase. During trot, the dog will be moving forward with relative ease at a constant “jogging” speed. During the “sprint” phase, the dog will be considered to sprint when it is galloping. Both rotatory and transverse gallops will be accepted. During a sprint, the dog will be moving forward as fast as it seems to be able to do so, with the aim of constantly increasing or maintaining the top speed that it can be motivated to achieve. HR and BL measurements will be taken before physical performance tests (baseline, BASE), after trotting for 1000 m (POST1), after sprinting for 200 m (POST2), and at recovery (REC) 5–8 min after the last run. These measurements will take about 10 min to complete. All measurements will be recorded on the test form (Appendix A).

Running distances will be measured with a rangefinder and marked clearly with cones. During the test, the dog will wear a non-slip collar or harness, and it will have water available after the test. Each dog will use their own individually fitted restraints (i.e., harness or collar and lead). Appropriate fit will be confirmed by the person responsible for testing. In the case of inappropriate gear, an individually adjustable collar will be provided by the research team.

The test will be interrupted if the dog becomes lame or shows any signs of excessive fatigue. Signs of low-level fatigue include, for example, sudden reluctance to move or excessive panting [35,37]. Signs of severe fatigue include dogs going awry or collapsing [35,37]. As the dogs in this study will be young, the test cannot be continued until severe signs are noticed, but the test must be discontinued at low-level signs. The signs will be observed by the owner as well as the veterinarian and the assistant (a veterinary nurse), all familiar with the breed.

The dogs will trot alongside their owner or an assistant for 1000 m. If a constant trot is maintained, the speed of the trot will not be controlled. After trotting for 1000 m, the dog will be stopped and POST1 measurements will be performed. Immediately, HR will be recorded from the pulse monitor’s mobile application (app), and blood from the inner side of the ear flap will be drawn to assess the BL level. HR will also be measured with a stethoscope if there is suspicion of sensor-to-app connection failure (noted as an exceptionally low or high HR or no HR). The test will continue immediately after the measurements with a 200 m sprint, running as fast as possible. After sprinting the 200 m, the dog will again be stopped, and HR and BL measurements will be repeated (POST 2). Once the POST2 measurements are conducted, the dog will be encouraged to lie down and have a rest, with no stimulus provided. This will be accomplished through verbal and non-verbal instructions and the dog’s motivation. The dog needs to lie down as it needs to calm down and be relaxed. Once the dog is calm, neither the owner nor the research personnel will speak to it. After 5–8 min, HR and BL will be measured again (REC). The HR is monitored through the app so that the dog will not be disturbed. HR will be monitored until it is back to the base value. The time taken to achieve the level of resting HR will be recorded and considered HRR. The running speeds of the 1000 m trot and 200 m sprint will be obtained from the app.

### 2.2. Description of Measurement Technique

All the fitness test measurements will be conducted immediately, which means within a few seconds after exercise. During the measurements, an assistant will hold the dog. In cases of behavioral reasons (i.e., a sensitive dog), the owner can handle the dog. In this case, owners will be instructed to handle the dogs in a standardized manner to ensure proper and consistent management of the dogs throughout the test and thus to verify the acceptable repeatability of the protocol. It will be allowed to stand or sit but not to lie down, except for the REC measurements, when the dog will be encouraged to lie down. The HR of the dog will be recorded by auscultation with a stethoscope placed on the left side of the dog at the fifth to sixth intercostal space. The heart will be auscultated for 15 s. The result obtained will be multiplied by four to obtain beats per minute (bpm). The result will be referred to as the baseline HR. Auscultation will be conducted prior to placing the HR sensor, which will be set to the same, above-described place. The HR sensor (Polar H10 sensor, Polar Electro Oy, Kempele, Finland, Cat. No.: 92075957) will be attached to its chest strap. To improve contact between the chest strap and the skin, the dog’s coat will be watered on the left side of the dog at the fifth to sixth intercostal space, and veterinary lubricant gel (Kruuse Bovivet Gel, Kruuse, Langeskov, Denmark) will be applied to the measurement area of the chest strap. It will then be attached to the dog’s chest. The correct place for the HR sensor will be confirmed by palpating the heart through the sensor. If needed, the dog is allowed to acclimate to the chest strap for up to 15 min. Prior to attaching the HR sensor to the dog, the sensor will be connected to an app (Polar Equine app, version 1.2.1, Polar Electro Oy, Kempele, Finland), which can be downloaded from Google Play (Google Ireland Limited, Dublin, Ireland) to a smartphone. For the app to work, the device must be an Android phone with an operating system of 6.0 or newer, and the phone must be at least Bluetooth 4.0 compatible. The app will display HR in real-time, so HR can be monitored from a distance without disturbing the dog. The app will also collect data on running speed and distance.

The effects of various actions during the test (such as taking blood samples for BL measurements) on the HR can be seen through the continuous collection of HR data. After attaching the HR sensor to the dog, the application is confirmed by auscultating the HR simultaneously.

While the owner or assistant holds the dog, the inner surface of the dog’s ear flap will be wiped with an antiseptic and allowed to dry for 5–10 s. If the ear flap is still wet, it can be wiped with a clean, dry gauze. A sterile 22-gauge needle will be used to make a small puncture on the inner surface of the ear flap. A blood drop will be drawn and applied immediately to a reactant strip placed on a handheld lactate analyzer (Lactate Scout 4, SensLab Gmbh, Leipzig, Germany, Cat. no.: 7023-0441-0246) to determine BL.

## 3. Expected Results

The sample size calculations will employ results of previously published similar studies [6,15,56,57,58] and required estimates based on clinical experience. A statistician specialized in medical research will be consulted in the assessment of the required number of dogs. Power analysis with at least 80% power will be used with a statistical significance level of *p* = 0.05.

Primary outcome variables of the test protocol will be changes in BL, HR, and HRR induced by an eight-week training intervention. Changes in these parameters will be evaluated by comparing measurements at baseline, after the intervention, and after the detraining period. Maintenance of the changes will be assessed by measuring the parameters at the end of the intervention and after a four-week detraining period. Descriptive statistics will be calculated for the measured HR, BL, and HRR values. For BL, change from POST2 at REC will also be calculated. The statistical calculations will be performed using SAS software version 9.4 (SAS Institute Inc., Cary, NC, USA).

## 4. Discussion

Although no tests are available for young dogs, some fitness tests for adult dogs have been published [58,59,61,62,63]. Most of the published fitness tests are laboratory types performed on a treadmill, or if they are actual field tests, they do not specifically measure endurance fitness but focus more on overall physical fitness [58,59]. One study has assessed the reliability of a treadmill exercise test against BL, HR, and body temperature responses to incremental exercise in dogs [61]. This test or its modifications have been used in other studies measuring endurance fitness in dogs [61,62,63]. However, treadmills may not be readily available to everyone needing to test dogs’ endurance fitness. Thus, tests without treadmills could be more feasible, although they might have their challenges. For example, in our study, as the puppies grow, their running speed during sprints might be too fast for a human to keep up with. Therefore, without a treadmill, the proposed protocol would not be suitable for adult dogs.

In our proposed protocol, the distances to be trotted and sprinted are chosen according to the best clinical knowledge from both human and veterinary medicine. In humans, there are no studies or unified opinions about a suitable running distance for children. According to Finley et al. [64], most of the children’s cross-country coaches consider a suitable running distance in competitions for 6- to 9-year-olds to be 800–1600 m. Based on clinical experience and common mutual understanding, a suitable running distance for young dogs is highly dependent on the breed, but a puppy of a non-chondrodystrophic breed should be able to run for at least 5 min per month of age, two times a day. In our protocol, trotting for 1000 m represents aerobic performance, and sprinting for 200 m represents anaerobic performance. A challenge of these running distances may be that they are too short to elevate a young dog’s HR and BL.

In humans, numeric tables are used in field tests to quantify endurance fitness [48,49,65,66]. These tables factor in the time used for the performance or distance covered in a certain time [48,49,65,66]. In horses, typically measured physical parameters in fitness field tests are HR, HRR, and BL [26,27,67,68]. These also apply to canine endurance tests performed on a treadmill, with the addition of body temperature as a typically measured variable [36,59,69,70]. In our proposed protocol, BL, HR, and HRR will be used to evaluate the changes in endurance fitness. We will not use body temperature because with young dogs it might take too much time to measure between the aerobic and anaerobic performances, giving the dogs too much rest between performance sets.

A specific HR sensor, originally designed for humans, has also been validated for dogs [20,21]. However, this particular device is no longer available, and therefore, the Polar H10 sensor will be used in this test. With this HR sensor and the Polar Equine app, HR as well as running times and speeds can be measured. Possible challenges may be that the HR sensor belt around the dog’s chest can disturb the dog while running. Also, if the dog has a very thick coat or is very obese, the contact of the HR sensor belt might be insufficient. If possible, hair could be shaved from the sensor area, but this is not obligatory. Although shaving hair would improve sensor contact, as the dogs included in the study will be privately owned, the decision on clipping the hair is the owner’s decision. Moreover, mandatory hair clipping might affect the owner’s willingness to use the HR sensor belt and thus the whole test.

There are only a few HRR-related studies in animals, and most of them have been conducted in horses [26,27,60]. In this study, we will use HRR as a performance indicator in endurance testing because it is a simple, inexpensive, and non-invasive method. The behavior of young dogs may vary markedly, and if the dog is very sensitive, it might also have an influence on the HRR.

For BL measurements, a handheld Lactate Scout 4 analyzer will be used in our study. Lactate Scout analyzer measurements have been shown to be comparable with laboratory methods [34,37]. The analyzer is simple, portable, and rapid, making it ideal for field use. In addition to the results being readable in 10 s, only a very small amount of blood (0.5 µL) is needed for the sample. The analyzer also has a wide range of measurements (0.5–25.0 mmol/L) (14). In the protocol proposed here, the blood sample will be taken from the inner surface of the ear flap, but if this is not possible, the sample can be taken from the metacarpal pad.

In human medicine, physiological responses to exercise are known to change during growth and development, and thus, pediatric tests have their own outcome reference values instead of sharing the adult norms [71,72]. The musculoskeletal system of young dogs is somewhat different from that of an adult dog. The biggest differences are seen in the functions of the heart and circulatory system. In young dogs, the heart size is larger in relation to the body weight than in adult dogs [73]. On average, growing dogs have lower red blood cell, hemoglobin, and hematocrit values than adult dogs, which weakens the transport of oxygen in the bloodstream [74,75]. Therefore, the results of fitness tests on young dogs are not comparable to those of adult dogs.

To the authors’ knowledge, there are no published articles about fitness testing in young dogs. The proposed protocol for field testing the endurance fitness of young dogs is useful because the equipment and environment needed for the test are inexpensive and readily available, and the protocol is non-invasive.

## Figures and Tables

**Figure 1 mps-06-00061-f001:**
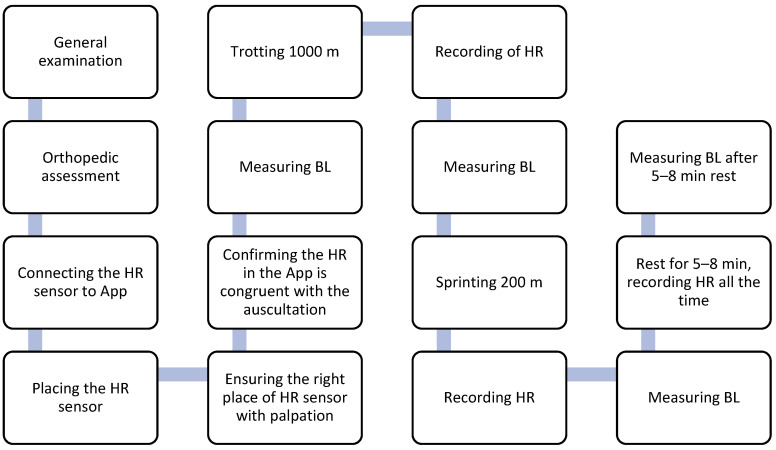
The structure of the endurance fitness test procedure. The procedure will be carried out in the same way for each test session. HR = heart rate, BL = blood lactate, App = Polar Equine App.

**Figure 2 mps-06-00061-f002:**
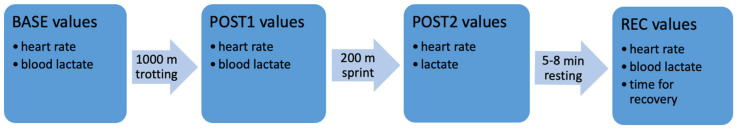
The structure of the fitness test session. Each of the fitness test sessions will consist of four measurements of heart rate (HR) and blood lactate (BL): baseline at rest (BASE), after trotting 1000 m (POST1), after sprinting 200 m (POST2), and recovery of 5–8 min after the last run (REC).

## Data Availability

No new data were created or analyzed in this study. Data sharing is not applicable to this article.

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
