# Peer review of "Proposed Protocol for Field Testing of Endurance Fitness of Young Labrador Retrievers"

_mps, 2023, doi:10.3390/mps6040061_

Round 1

Reviewer 1 Report

Researchers are correct that there is limited research concerning exercise testing utilizing the canine population, and yet, the performance canine industry is growing along with the participation within recreational activities for dog owners such as biking, running, and walking with dogs is on an upward trend. Thus, this manuscript is relevant to the current industry.

As for the content of this manuscript, the bulk of the introduction is well organized and easy to follow with clear definitions for those unfamiliar with exercise physiology terminology. Nonetheless, the last paragraph of the introduction needs to be strengthen with discussion of research concerning the recent rise in interest in dog sporting and other physical activities and the potential injury due to improper training and conditioning. Improper training and conditioning studies can be that found in equine, although there are some case studies and survey data reflecting potential risks related to some of the physical activities associated with canine use. This discussion concerning the rise of physical activities within the canine industry and the potential risk of injury without proper training is the justification of this study, and thus, should be a larger percentage of the introduction. 

As for the procedure section, section 2, discuss in detail your recruitment process of participants. If intended number is 16 canine, justify as to why, and include details of the type of animals within that 16 researchers will be recruiting such as specific age range, size, and breeds. Include specifics on the inclusion and exclusion criteria to be utilized. Part of this criteria needs to include specifics about the clinical examinations that will be done prior to the protocol and specifics on the details on health history that will be required. What specific genetic testing will be required and include justification. What will be the timing as it relates to the clinical examination including timing of diagnostic testing. Give justification for the inclusion/exclusion criteria, number and type of animals, and the timing of the examination and testing to be done. Discuss power analysis concerning the numbers that will be utilized and estimates concerning retention and consent rates. 

How will the dogs be divided into groups? If randomized, what statistical procedure will be done to ensure randomization? In line 103, define "normal puppy life" along with details concerning restrictions when saying "without restrictions or added exercises". Give justification on the weeks and timing concerning the exercise protocol for the treatment group discussed starting in line 100-106 and 113-119. As for environment, give justification for why the type of environment for this protocol and the length for acclimation as described in lines 108-112.

Within lines 120-125, give references as to what research documents the signs that will be utilized for fatigue for the canine and specifics as to what will happen if that occurs beyond the test being "interrupted". In fact, a section focusing on safety, adverse events, and potential participant withdrawal should be included, especially since the researchers plan to use owners for handling the dogs during the trials (See below for further comments concerning owner participation). Specifically, how will safety of dog and owner participants be ensured, how will adverse events be documented and adjusted for, what will be included in adverse events, what are the withdrawal procedures for dog and/or owner participant, can the researchers mandate a withdraw of dog and/or owner and if so how, and how will researchers adapt and document for such withdraws? 

Give details on the type of collar/harness to be utilized and how that will be ensured for proper fit so that it does not interrupt canine performance. As mentioned previously within this review, protocol indicates, starting on line 126, that either owner or assistant will handle canines during protocol, however, further details are needed on how the handler of the dogs during this protocol will be properly and consistently handled. Researchers not only want to ensure humane handling, but also consistency so that the protocol can be repeatable. Give justification including referencing for the type of training given. It is advisable to utilize a veterinary assistant instead of owner as owner participation would require additional consent in case the owner was hurt during participation within the study. Veterinary assistant will also be more knowledgeable in detecting potential early signs of lameness and/or fatigue. Nonetheless, if owner is to be an option for this protocol, the discussion concerning consent needs to cover both for the use of the dog as a research animal and the participation of the owner as a part of the research protocol including waiver of consent for injury of both dog and owner.  

The jog and trot within the procedure section are utilized interchangeably, but since these are two types of gaits, utilize only one term. See additional remarks concerning gait within this review as it pertains to the discussion section. Define objectively the gait utilized for this protocol and describe how that gait will be ensured it is being performed constantly as mentioned in line 126 if a treadmill is not being utilized? Why will speed not be controlled as speed selection influences stride variables and these variables such as the use of suspension compared to stride frequency can account for fatigue. Changes in speed can indicate not only fatigue but potentially lameness. 

Starting in line 126, timing of measurements is discussed, but exact timeframe is limited to just saying "immediately", however, to ensure repeatability specifics need to be given. Again, justify utilizing previous references as to the distances utilized. After POST2, dogs will be "encouraged to lie down and have a rest", but how will this be accomplished and why is lying down needed? Exactly describe the resting period and what that will consist of for the dog including how the researchers will ensure the dog will not be disturbed (lines 135-137). If a dog is unwilling to lie down will that lead to mandated withdraw, and if not, how will differences in stance of the dog during this time period be accounted for?

For the HR sensor, the coat at the site will need to be cut so that the haircoat does not deter measurements nor allow for displacement of sensor. See further comments within this review within the discussion section concerning these concerns. Utilize references to support your procedures concerning HR measurements. Discuss fit to ensure it does not impede performance and discuss acclimatization period after sensor is attached as this will be important to ensure the novelty of the sensor does not promote changes in HR. Furthermore, how will blood draws be ensured not to cause rise in HR? 

Remove section "2.3. Figures" and move figure with it's respective title to after line 119 at the end of the paragraph where Figure 1 is mentioned. Relocating figure is helpful for readers to visualize the components of the fitness test as it is being discussed. The section on expected results needs to be expanded to include further details concerning potential outcomes for each parameter being measured including potential unforeseen results. Each parameter needs to be discussed separately along with correlations potentially made between the parameters and trends observed. Recognize that there are two groups, and thus, outcomes should include this potential comparison. Finally, these outcomes are dependent on adequate statistical analysis so give further details on the specific analysis procedures that will be applied. Again, keep in mind there are two groups along with multiple measures taken where there may correlations. Will the two groups be balanced? What is the hypothesis that you are testing? Discuss accordingly.

As mentioned previously, the lack of controlling for speed can be a potential problem, and although the use of a treadmill will assist with this issue and assist with the issue brought up concerning the handlers during the study (see previous comment on owners handling dogs during protocol), the researchers discuss in lines 189-196 the benefits of the lack of a use of a treadmill for the proposed protocol. The disadvantages of the use of a treadmill, particularly as it pertains to gait mechanics, should be explored in more detail and discussion on how this protocol will account for these disadvantages must be included. In lines 198-208, the terms used concerning gaits seem to be interchangeable, although as mentioned previously, there are clear differences between gaits that will influence variables measured. Ensure the studies discussed within this discussion section include the correct gaits mentioned in that respective study and make sure they are applied accordingly to the current protocol. There unique biomechanical variables between jog and trot, sprint and run, etc. along with differences between bipeds and quadrupeds and between species. Further, keep in mind that gait is very dependent on the breed selected for this protocol and that includes the speed along with other temporal variables performed during the gait, thus, this needs to be addressed and accounted for within this discussion. This may also indicate that without a mandated treadmill for the protocol that the protocol should be set to a specific breed and/or size of dog so that the protocol isn't applied to a larger dog breed where the handler cannot properly track the same distances as the dog's natural stride, and so, impeding natural performance of the gait. This would require adjustment in the title to include not only young dogs, but the type of breed this protocol is suitable for. As seen even within this discussion section as given by the researchers, the protocol distances are dependent on breed type. If protocol is not adjusted to a specific breed, then, this section within the discussion needs to be further expanded to include how these limitations will be accounted for.

In lines 217-218, the justification for not using temperatures is that the puppies may be "afraid" of taking temperatures, but wouldn't blood draws also elicit fear responses? There are also non-invasive ways today for measuring temperatures beyond rectal temperatures. Further support beyond fear should be given for why temperatures would not be included within this protocol. Again, for line 225, coat will need to be shaved to ensure for proper fit and proper measurement. For line 230, use more objective, scientific terminology besides "shy" and "timid" to better quantify behavioral responses. Include references that support this conclusion concerning these responses as it relates to HR and how this will be accounted for within the protocol. Finally, last paragraph needs to be put into a separate section labeled as "5. Conclusions".

See previous comments for authors.

Reviewer 2 Report

With regard to manuscript: Proposed protocol for field testing of endurance fitness of young dogs, for consideration in Methods Protoc. This is a very interesting manuscript proposing a protocol for testing of endurance fitness of young dogs. The paper is worthy of publication, however, I have to be honest in mentioning the absence of preliminary results demonstrating the validity of protocols. With the utmost respect, allow me to give you a few suggestions.

·  The novelty of the study could be more highlighted (in the introduction). The authors would explain why their findings aggregate to the existing knowledge.

·  Introduction: The explanation about field test batteries in dogs was vague.

·  Introduction: The aerobic threshold (in km/h) for endurance-trained dogs could be defined in the introduction to facilitate the compression of the reviewer/reader. This information can be useful.  

·  Didactics would improve with the inclusion of pictures showing the timeline and details of protocols.

·  The authors do not explain the reason why they decided to use heart hate (why not running efficiency by video analyses, and others).

·  About data collection, give more details on the place of evaluations, time of day, time in each individual evaluation and other minor things.

·  It is unclear why a pilot study was not shown. A limitation of this study is the lack of preliminary findings data.

·  Young dogs differ significantly from mature dogs in several aspects of body structure (body composition). This should be highlighted in our proposed study. I would like this point of view to be more in-depth in discussion.

·  It would be interesting to know about the energy intake of dogs. This practice would have any influence on their findings.

Round 2

Reviewer 1 Report

Authors are commended for a thorough revision of the manuscript. Some revisions recommended by the reviewer were not addressed due to the limitations of the journal, but overall, most were addressed within the updated manuscript. A few minor revisions are still needed, however. Remove sentence within lines 121-124 as it can include bias to your study to have owners make decisions and a coin toss is not typical for scientific methods. Instead, just include a statement suggesting assignment of groups will be randomized. Similar removal is warranted for the sentence within lines 138-139 as the time of day for testing can influence results, and thus, instead indicate that time of day will remain consistent throughout the study. Additionally, remove the sentence within line 143 concerning not utilizing diagnostic imaging. Instead, just indicate what assessment methods will be utilized by the veterinarian such as physical assessment including palpations and observation of gait for indication of unsoundness. Finally, within the methods section, and specifically within the paragraph starting in line 164, include a definition of the trot and sprint. This would be a part of the inclusion criteria that the canine is properly performing the gait as improper gait can influence results. Thus, a definition of what would be considered correct gait needs to be included clearly outlining the specific mechanics of the gait to ensure the canine isn't walking when it should be trotting and/or trotting instead of sprinting. Utilization of breed standards for the Labrador Retriever would be helpful for gait descriptions. 

Author Response

Remove sentence within lines 121-124 as it can include bias to your study to have owners make decisions and a coin toss is not typical for scientific methods. Instead, just include a statement suggesting assignment of groups will be randomized.

  • Thank you for pointing this out. We did as you suggested and removed the sentence from original lines 121-124. According to your instructions, it has now been replaced with a sentence “The dogs will be randomized to these groups” (see lines 121-122).

Similar removal is warranted for the sentence within lines 138-139 as the time of day for testing can influence results, and thus, instead indicate that time of day will remain consistent throughout the study.

  • We fully agree with this point and have thus removed the text from original lines 138-139 and added instead “remains consistent throughout the study” on line 136.

Additionally, remove the sentence within line 143 concerning not utilizing diagnostic imaging. Instead, just indicate what assessment methods will be utilized by the veterinarian such as physical assessment including palpations and observation of gait for indication of unsoundness.

  • According to your suggestion, this sentence has now been replaced by “lameness evaluation will be done by palpation and manipulation as well as  gait observation for any indications of unsoundness” (lines 140-141).

Finally, within the methods section, and specifically within the paragraph starting in line 164, include a definition of the trot and sprint. This would be a part of the inclusion criteria that the canine is properly performing the gait as improper gait can influence results. Thus, a definition of what would be considered correct gait needs to be included clearly outlining the specific mechanics of the gait to ensure the canine isn't walking when it should be trotting and/or trotting instead of sprinting. Utilization of breed standards for the Labrador Retriever would be helpful for gait descriptions.

  • Thank you for your advice. We added definitions of trot and sprint to lines 144-150: “During “trot” phase, running trot will be utilized, defined with a running gait with a suspension phase between each diagonal support phase. During trot, the dog will be moving forward at a relative ease, constant “jogging” speed. During the “sprint” phase, the dog will be considered to sprint when it is galloping. Both rotatory as well as transverse gallop will be accepted. During sprint, the dog is to move forward as fast as it seems to be able to do so, with the aim of constantly increasing or maintaining the top speed that it can be motivated to achieve.“

Reviewer 2 Report

My recommendation is to accept the manuscript. Huge improvements have been made in terms of advances and understanding of the proposed.

Author Response

My recommendation is to accept the manuscript. Huge improvements have been made in terms of advances and understanding of the proposed.

- Thank you for your kind comment.